# The Expression of the Immunoproteasome Subunit PSMB9 Is Related to Distinct Molecular Subtypes of Uterine Leiomyosarcoma

**DOI:** 10.3390/cancers14205007

**Published:** 2022-10-13

**Authors:** Raul Maia Falcão, Georgia Kokaraki, Wout De Wispelaere, Frédéric Amant, Gustavo Antônio De Souza, Jorge Estefano Santana de Souza, Joseph Woodward Carlson, Tirzah Braz Petta

**Affiliations:** 1Bioinformatics Graduate Program, Instituto Metropole Digital, Federal University of Rio Grande do Norte, Natal 59078-970, Brazil; 2University of Southern California Keck School of Medicine, University of Southern California, Los Angeles, CA 90007, USA; 3Department of Oncology, Leuven and Leuven Cancer Institute, Katholieke Universiteit, 3000 Leuven, Belgium; 4Department of Surgery, The Netherlands Cancer Institute, 1066 Amsterdam, The Netherlands; 5K7 Onkologi-Patologi, Karolinska Institute, 17177 Stockholm, Sweden

**Keywords:** uterine leiomyosarcoma, immunoproteasome, PSMB9, extracellular matrix

## Abstract

**Simple Summary:**

Uterine leiomyosarcoma (uLMS) is a rare, aggressive, and highly heterogeneous tumor. Knockout female mice for the catalytic subunit of the immunoproteasome PSMB9 develops spontaneous uLMS. In this study, we used molecular data from 3 non-related uLMS cohorts that were integrated and analyzed by proteotranscriptomics. We observed overexpression of the immunoproteasome pathway in uLMS, and then further classified the samples as low or high PSMB9 gene expression levels and we provide evidence that; (i) in the group high there is an enrichment of pathways related to the immune system and in the group low, the ECM formation; (ii) samples with high CD8+/PSMB9 ratio shows better OS; and (iii) the main regulator in the high group is IFNγ and in the low, the proto-oncogene SRC. These findings contribute to the understanding of potential therapeutic or prognostic markers in uLMS.

**Abstract:**

Background: Uterine leiomyosarcoma (uLMS) are rare and malignant tumors that arise in the myometrium cells and whose diagnosis is based on histopathological features. Identifying diagnostic biomarkers for uLMS is a challenge due to molecular heterogeneity and the scarcity of samples. In vivo and in vitro models for uLMS are urgently needed. Knockout female mice for the catalytic subunit of the immunoproteasome PSMB9 (MIM:177045) develop spontaneous uLMS. This study aimed to analyze the role of PSMB9 in uLMS tumorigenesis and patient outcome. Methods: Molecular data from 3 non-related uLMS cohorts were integrated and analyzed by proteotranscriptomic using gene expression and protein abundance levels in 68 normal adjacent myometrium (MM), 66 uterine leiomyoma (LM), and 67 uLMS. Results: the immunoproteasome pathway is upregulated and the gene PMSB9 shows heterogeneous expression values in uLMS. Quartile group analysis showed no significant difference between groups high and low *PSMB9* expression groups at 3-years overall survival (OS). Using CYBERSORTx analysis we observed 9 out of 17 samples in the high group clustering together due to high M2 macrophages and CD4 memory resting, and high CD8+/PSMB9 ratio was associated with better OS. The main pathway regulated in the high group is IFNγ and in the low is the ECM pathway dependent on the proto-oncogene SRC. Conclusion: these findings suggest 2 subtypes of uLMS (immune-related and ECM-related) with different candidate mechanisms of malignancy.

## 1. Introduction

Uterine Leiomyosarcoma (uLMS) is a soft tissue sarcoma (STS) that arises in the cells of the myometrium [1]. Furthermore, uLMS is a rare and aggressive malignant tumor with an estimated annual incidence of 0.55–0.92 per 100,000 women. It represents the most common type of uterine sarcoma with an aggressive clinical course [2]. A study based on Surveillance, Epidemiology, and End Results (SEER) indicates that poor survival was associated with old age, black race, and advanced disease stage [3]. According to the SEER, the 5-year Relative Survival Rate of women diagnosed with uterine sarcoma between 2010 and 2016 was 66%, 34%, and 13% for the primary site, regional, and metastasis, respectively. uLMS diagnosis typically occurs by chance when performing hysterectomy for leiomyoma (LM) and is confirmed by histopathological characteristics such as cell atypia, mitotic index, and tumor cell coagulative necrosis [4,5]. Although gynecological cancers are frequently associated with female hormones, no correlation has been reported between hormone receptors and survival rates in patients with uLMS. Furthermore, no standard effective type of chemotherapy or radiation therapy has been identified for uLMS and surgical intervention remains the mainstay of treatment.

Identifying diagnostic biomarkers for uLMS is still a challenge due to its molecular heterogeneity and scarcity of samples. This molecular heterogeneity can contribute to resistance to treatment by radiotherapy and chemotherapy, thus identifying new therapeutic targets for uLMS is required for a more personalized form of medicine and to improve both patient survival rates and quality of life [6].

It has been shown that the absence of PSMB9 is associated with the development of spontaneous uLMS, where ~36% of the homozygous PSMB9-deficient mice developed the tumor by 12 months of age [7], and the malignancy of uLMS driven by the deficiency of PSMB9 is directly correlated to the Interferon Regulatory Factor 1 (IRF1) expression. Also, it was shown that PSMB9 expression is absent in human uLMS but present in LM suggesting PSMB9 is a potential diagnostic biomarker and tumor suppressor for uLMS.

The gene PSMB9 (MIM:177045), also known as LMP2, encodes a catalytic subunit of the immunoproteasome which replaces the standard catalytic subunit in the 20S core during the proteasome assembly [8]. Among immunoproteasome functions, protein degradation is a major role in the ubiquitin-proteasome system leading to a generation of ligands for major histocompatibility complex (MHC) class I antigen presentation and also the regulation of CD8+ T cells [9,10,11]. As an example, mutant mice deficient in PSMB9 were found to have a reduced level of CD8+ T cells and a reduced response to the nucleoprotein epitope of influenza A [12]. *PSMB9* is an interferon-gamma (IFN-γ) inducible factor and the gene is located within the MHC class II region in chromosome 6 [13]. Also, *PSMB9* is directly regulated by anti-oncogenic IRF1, a well-known gene with suppressor tumor activity [14,15,16,17].

In this article, we integrated molecular data from different sources of tumors to investigate PSMB9 expression and its relationship with uLMS patient prognosis. We have dichotomized the samples in low and high PSMB9 expression levels and performed differential expression analysis followed by Gene Set Enrichment Analysis (GSEA). We found that samples with high PSMB9 expression levels present activation of pathways related to IFN-γ and the immune system, while pathways related to the development and formation of extracellular matrix (ECM) were suppressed. The high PSMB9 expression level group showed a better prognosis when considering the CD8+/PSMB9 ratio at a 3-year survival point, indicating that PSMB9 expression level may be associated with immune response in uLMS. Finally, the prediction of low PSMB9 protein abundance showed inhibition of pathways related to the development and formation of ECM through the proto-oncogene tyrosine-protein kinase (SRC) gene.

## 2. Materials and Methods

### 2.1. Patient Cohort

A total of 68 MM, 66 LM, and 67 uLMS samples were analyzed by transcriptomics and the data were normalized and integrated (Appendix AA). Also, proteomics data consisted of a total of 10 MM, 11 LM and 17 uLMS samples to be normalized and integrated with transcriptomics results. RNA sequencing quantified 19,291 protein-coding genes while the proteomics data quantified 9470 proteins. The posterior RNA-Seq filtering step in the count table resulted in 14,045 protein-coding genes. We observed a batch-effect due to library size among different projects (Appendix AB) and it was corrected after TPM normalization (Appendix AA–C) resulting in an R-squared (R**^2^**) value equal to 0.798 (MM × uLMS), 0.814 (LM × uLMS) and 0.941 (MM × LM).

### 2.2. RNA-Seq Data Collection

A cohort of 95 female patients who have been submitted to surgery at the Karolinska Hospital from the Karolinska Institute Hospital, Sweden, between the years 2002 and 2021. The protocol of the present study was approved by the Ethics Review Board of the Stockholm Region (DNR 2015-143-1) and all methods were performed in accordance with relevant guidelines and regulations and in compliance with the principles outlined in the Declaration of Helsinki. In total, 124 fresh frozen tissues were collected. Of those, 93 samples were submitted to the RNA-Seq experiment and 58 samples were submitted to generate protein abundance data. Some of those samples had both RNA-Seq and proteomics experiments (nMM = 7, nLM = 7 and nuLMS = 15). Publicly datasets were obtained from NCBI Sequence Read Archive (SRA; https://www.ncbi.nlm.nih.gov/sra, accessed on 15 February 2021 using the following search terms: “Uterine Leiomyosarcoma’’, “Leiomyoma’’ and “Fibroid”. We found five projects (PRJEB21003, PRJNA391373, PRJNA498292, PRJNA526865, PRJNA558981) with fresh-frozen patient tissue samples from the RNA-Seq experiment. From those, only the PRJEB21003 project included uLMS e MM samples. The remaining projects included LM and MM samples. All data were downloaded using the parallel-fastq-dump (https://github.com/rvalieris/parallel-fastq-dump tool accessed on 20 February 2021. Also, clinical data and controlled access bam files were obtained from The Cancer Genome Atlas (TCGA) Sarcoma cohort (TCGA-SARC). A total of 24 samples were retrieved from female gynecological and uLMS as a histologic diagnosis. Samples from tumors outside the uterus were excluded.

### 2.3. RNA and Protein Extraction and Library Preparation

Tissue blocks were sectioned into 10 μm slices using a cryostat at −20 °C and seven consecutive sections were pooled and used as one sample for RNA and protein extraction and stored at −80 °C until analysis. AllPrep DNA/RNA/Protein Mini Kit (QIAGEN, Hilden, Germany) was used for the extraction as described in the manufacturer’s protocol. RNA libraries were prepared using the TruSeq Stranded Total RNA with Ribo-Zero Protocol sample preparation kit v2 (Illumina, San Diego, CA, USA) following the manufacturer’s instructions. Sequencing of the libraries (2 × 150 reads) was performed by the NOVASeq6000 S4 platform (Illumina).

For the generation of protein abundance data, the high-performance 16-plex TMT-LC/LC-MS/MS protocol was used for proteomic profiles. Samples were prepared for mass spectrometry analysis using a modified version of the SP3 protein clean-up and digestion protocol. Peptides were labeled with TMTpro 16 plex reagent according to the manufacturer’s protocol (Thermo Scientific, Waltham, MA, USA).

Controlled access bam files from TCGA were downloaded through gdc-client (v1.6.1) tool. Then we converted back to fastq files using the fastq function from samtools (v1.9). Next, we sorted the paired-end reads using fastq-sort from fastq-tools (v.0.8.3). Finally, the same pipeline described below was used for all samples.

Each fastq file was assessed using FastQC (v0.11.9) followed by adapter removing and trimming bad quality base calling with Trim-galore (v0.6.6) when it was detected in the quality report. The GRCh38.p13 human genome reference and genome annotation were downloaded (https://ftp.ncbi.nlm.nih.gov/refseq/H_sapiens/annotation/annotation_releases/109.20210226/GCF_000001405.39_GRCh38.p13/, accessed on 1 May 2021). Genome reference was indexed with HISAT2 (v2.1.0) using Hierarchical Graph FM index (HGFM). At this step, only mRNA with a “protein-coding” biotype in a complete genomic molecule (RefSeq NC_format) and transcripts tagged as BestRefSeq were considered. Then, reads were aligned with HISAT2 using the parameter rna-strandness of each sample after inferring the experiment using the script infer_experiment.py from RSeQC (v4.0.0). Samtools were used to filter out reads unmapped, supplementary alignments, and reads failing in platform/vendor quality checks and reads with mapping quality below 30 (F = 2828 and q = 30). Counting reads for each gene was performed using htseq-count (0.13.5) with union mode and the strand-specific information corresponding to each sample.

For protein analysis, we used MSstatsTMT (v2.4.1) to normalize the peptide-level data using the default settings and then performed the differential abundance analysis. The Benjamini-Hochberg procedure was used to control the FDR of *t*-test.

### 2.4. Differential Expression and Enrichment Analysis

We used DESeq2 (v1.36.0) to obtain log2FoldChange values in the High × Low group comparison. And to account for batch effects, we included BioProjects accession number and groups (MM, LM, uLMS) as a covariant in the DESeq2 design formulas. We chose ranking by log2FoldChange values from DeSeq2 and then the ranked genes were submitted to Gene Set Enrichment Analysis (GSEA) with fgsea (v1.22.0) using parameters such as nPerm = 100,000 and Reactome collection as input to be enriched. Top20 enriched pathways with |NES| ≥ 1 and *p*-value < 0.05 was displayed.

Gene sets from proteasome and interferon-gamma pathways were retrieved using KEGG and Reactome collections, respectively, from the molecular signatures database (MSigDB). Hierarchical clustering was performed and shown as a heatmap using ComplexHeatmap (v2.12.1). Row Z-score was calculated based on log2(TPM + 1) values.

Within uLMS samples, quartile rank was identified using fabricatr (v1.0.0) and samples were divided into two groups according to PSMB9 expression: low expression level (with TPM values in the 1st quartile rank) and high expression level (with TPM values in the 4th quartile rank). RNA-Seq samples from our cohort found in the 1st or 4th quartile was mapped to our proteomics data to be labeled as belonging to the corresponding level group.

### 2.5. Ingenuity Pathway Analysis

The Molecule Activity Predictor (MAP) tool from Ingenuity Pathway Analysis (IPA) (QIAGEN Bioinformatics) was used to predict effect consequences due to activation or inhibition of PSMB9. To simulate the activation of PSMB9 we used the ratios from differential gene expression between High × Low group comparison using pathways related to the immune system. And, to simulate the inhibition of PSMB9 we used the ratios from differential protein abundance between High × Low group comparison using pathways related to the extracellular matrix.

### 2.6. Data Integration

Removing low expressed genes was done per project. Genes were filtered out if less than 50% of each group (MM, LM, uLMS) had count reads below 5. Then, genes neither present in all MM projects (constitutive genes) nor in uLMS or LM groups were removed. Next, expression values were calculated as Transcripts Per Million (TPM) using scater (v1.24.0) in order to remove library dependencies (i.e., a total of sequenced reads). Finally, each project was merged into one expression data table.

### 2.7. Cell Line Expression

TPM gene expression of 36 cell lines from female samples with primary disease sites in the uterus was downloaded at DepMap portal (https://depmap.org/portal/download/all/, accessed on 10 October 2021). Due to some nomenclature conflicts on disease classification, we double-checked this information on Expasy: Swiss Bioinformatics Resource Portal.

### 2.8. Immune Cell Subtype Related to Immune Activity

CIBERSORTx [18] was used to estimate absolute immune cells subtype fractions using gene expression data from high and low PSMB9 expression groups. LM22 signature gene file, 1000 permutations, quantile normalization disabled, and batch correction enabled were used as parameters to run CIBERSORTx. Values to measure immune activity were created using absolute cell fractions from tumor-associated macrophages (M1 and M2 macrophages) and CD8+ T cells. Ratios M1/M2 and CD8+/PSMB9 were calculated and considered as a metric to investigate the impact of those ratios on survival and to evaluate immune activity in uLMS samples. Both M1/M2 and CD8+/PSMB9 groups levels were defined using the median of ratio value: low (below the median) and high (above the median).

### 2.9. Survival and Statistical Analysis

Only TCGA-uLMS and our data had available clinical data containing overall survival (OS) time and patient status (dead vs. alive). Kaplan-Meier survival analysis was done using survival and survminer (v0.4.9) to calculate overall survival probabilities and to draw survival curves. Pearson correlation test was done using stats (v4.2.1) and correlation plot was performed using corrplot (v0.92). All the analysis described above was done using only uLMS samples from 1st and 4th quartiles. The Kaplan-Meier curve generated using all samples from TCGA-SARC was done at The University of ALabama at Birmingham CANcer data analysis Portal (UALCAN) (http://ualcan.path.uab.edu/, accessed on 4 September 2021).

## 3. Results

### 3.1. PSMB9 and the Proteasome Pathway Are Highly Expressed in uLMS, but Not in LM

We used a unique cohort from tumors analyzed by our group pooled with a public database with cases of uLMS and LM resulting in a total of 68 MM, 66 LM, and 67 uLMS (Appendix A). After the data were normalized, we compared the expression level of PSMB9 in each group (Appendix AA). Our analysis shows higher PSMB9 expression levels in uLMS compared to the benign tumor, LM, and normal adjacent myometrium, MM groups (Figure 1a), but no difference was observed at the protein level (Figure 1b) nor in gynecological cell lines (uterine leiomyosarcoma, endometrial/uterine cancer, and vulvar/ovarian leiomyosarcoma cell lines) (Figure 1c).

Unsupervised clustering to identify genes that are commonly regulated in the 3 groups (MM, LM and uLMS) showed the proteasome pathway is mainly overexpressed in the uLMS group when compared to the benign tumor (LM) and normal adjacent myometrium (MM) (Figure 2a). Immunoproteasome expression is known to be closely associated with IFNγ or T-cell infiltration [19], however the expression of the genes in the IFNγ pathway could not cluster the samples intragroup (Figure 2b).

### 3.2. PSMB9 Expression Level Groups Low or High Are Enriched for ECM and IFNγ Pathways, Respectively

For further analysis on the impact of PSMB9 expression level in uLMS tumor biology, we separated samples in the 1st and 4th quartiles. We observed 17 samples in each group. Gene set enrichment analysis to find out biological process associated with differentially expressed genes in each of the quartiles samples showed samples with high PSMB9 expression levels present the pathways “IFNγ signaling” and “immune system” activated, while “collagen formation” and “extracellular matrix (ECM) organization” pathways were suppressed (Figure 3). The opposite was found in the samples from the group low (data not shown).

To extrapolate activated/repressed pathways we aimed to identify joint differential expression in gene-pathway pairs through a correlation analysis between genes that are regulated by IFNγ signal cascade and immunoproteasome pathway. The expression of IRF1 and PSMB9 expression highly correlated (Pearson r = 0.79, *p*-value = 5.43 × 10^−25^) (Figure 4), while STAT1 and PSMB9 showed a moderate positive correlation (Pearson r = 0.64, *p*-value = 6.10 × 10^−13^). The transcription factors STAT1 and IRF1 present a moderate positive correlation (Pearson r = 0.68, *p*-value = 3.42 × 10^−21^), as both act to activate transcription of the PSMB9 gene.

### 3.3. Overexpression of the Immunoproteasome PSMB9 Subunit Is Correlated with uLMS Patients’ Survival Outcome

Immunoproteasome expression is known to be closely associated with IFNγ or T-cell infiltration, but it remains unclear whether expression levels of immunoproteasome subunits independently contribute to patient survival. We used CIBERSORTx to examine the absolute fractions of 22 infiltrating immune cell types in each tumor tissue. We observed 9 out 17 samples with high PSMB9 expression levels clustering together due to high cell fractions related to M2 macrophages and CD4 memory resting (Figure 5)

To assess the relationship between the expression levels of the immunoproteasome subunit PSMB9 with M1/M2 ratio and CD8+ in uLMS survival rate, we analyzed data from TCGA and our cohort patients for whom RNA-seq data and patient outcomes were available. Survival analysis showed no significant association between PSMB9 expression level and positive prognostic outcome (*p*-value = 0.56) (Figure 6a). However, an overall survival (OS) analysis using all samples from the TCGA-SARC showed a significant positive association between PSMB9 expression level and OS (Appendix A). This contrast can be explained by (i) in the TCGA-SARC there are 206 STS, among those, only 27 are uLMS; (ii) the small number of samples in our analysis with follow-up clinical data.

The M1/M2 ratio in PSMB low/high groups showed no association with a favorable outcome for in 3-year OS (*p*-value = 0.21) (Figure 6b). However, our analysis showed that the CD8+/PSMB9 ratio is associated with a better 3-year OS (*p*-value = 0.032) (Figure 6c).

Furthermore, CD8+ and PSMB9 had a low positive correlation (r = 0.46, *p*-value = 8.94 × 10^−5^) with 83% (5 out of 6) of samples in the low PSMB9 expression level group had low CD8/PSMB9 ratio, and 71% (5 out of 7) of samples in the high PSMB9 expression level group had high CD8/PSMB9 ratio (Appendix A).

Predicted consequences due to increased expression of PSMB9 identified two activated pathways, namely “antigen presentation” and “development of the immune system”, that play key roles in immune response (Figure 7a). Both pathways were indirectly activated through IFNγ. In turn, three pathways, namely the “development of extracellular matrix”, the “developmental process of extracellular matrix”, and the “formation of extracellular matrix”, were inhibited through SRC, a known proto-oncogene that plays a central role in the tumor development, growth, progression, and metastasis. Those inhibitions were influenced by the decreased abundance of PSMB9 through SRC (Figure 7b).

## 4. Discussion

In this study we proposed to understand the molecular mechanisms related to the gene *PSMB9* in the tumorigenesis of uLMS. For this, we have performed a deep integrated molecular analysis of a comprehensive annotated data set. Here, we provide evidence that the *PSMB9* gene is highly expressed in uLMS compared to MM and LM. Further analysis into categorized low/high *PSMB9* gene expression levels shows that (i) gene set enrichment is related to pathways in the immune system and ECM; (ii) CD8+/PSMB9 ratio shows survival impact in uLMS patients; and (iii) the main pathway regulated in the presence of PSBM9 high levels is related to the immune system and in low levels of *PSMB9*, the proto-oncogene *SRC* presents a central role.

Several studies provide evidence using gene expression profiling experiments, that uLMS has an overexpression in pathways related to cell cycle, the ubiquitin-proteasome system, and in genes with functional annotation related to muscle contraction [20,21,22,23,24,25]. The Ubiquitin-proteasome system plays an essential role in protein degradation and cellular processes, and because of those biological roles immunoproteasome has been studied in cancer due to its influences on immune surveillance and immune escape [26].

It has been proposed that deficiency in the gene PSMB9 could spontaneously induce uLMS in female mice. This occurs with a disease prevalence of ~36% by 12 months of age with reduced expression of IRF1 suggesting defective PSMB9 expression may be one of the causes of uLMS [27]. The spontaneous development of uLMS was observed due to a targeted disruption of PSMB9 proteasome subunits using gene encoding neomycin resistance (neo) driven by the PGK1 promoter [28,29]. It is known that this (neo) gene induces changes in the cells decreasing levels of mRNA in procollagen lα and fibronectin by 80% and 50% respectively both related to the ECM. This fact led us to hypothesize that the down-regulation or loss of expression of PSMB9 may characterize a subtype of uLMS related to ECM.

In this study, we demonstrated an upregulation profile in the KEGG proteasome pathway in the uLMS group compared to both LM and MM. Moreover, significant intra-tumoral differences were found in PSMB9, suggesting different subtypes within the uLMS group. Active pathways related to IFN-γ and the immune system suggest that proteasome upregulation is an indicator of an effective antigen presentation in the high PSMB9 expression level group. As described previously, the regulation of both adaptive and innate immune systems is mediated by the IFN-γ signaling cascade [30]. When Signal Transducer and Activator of Transcription (STAT1) is phosphorylated it induces the transcription of IRF1 and in its turn activates immunoproteasome subunits. Our data showed STAT1, IRF1, and PSMB9 were up-regulated in the high PSMB9 expression level group with Pearson correlation being stronger between STAT1 and IRF1 than between STAT1 and PSMB9, thus suggesting the sequential order in IFN-γ signaling cascade.

Although no significant association was found in the M1/M2 ratio, the presence of CD8+ tumor-infiltrating lymphocytes (TILs) in the high group, the ratio CD8+/PSMB9 showed a stronger association with OS in uLMS cell type.

We have used the proteomics data to analyze the main regulators in the ECM pathway since ECM proteins are synthesized and processed in the endoplasmic reticulum and then exported. PSMB9 and SRC down-regulation inhibits the ECM pathway, but even when PSMB9 is up-regulated and SRC is down-regulated the ECM pathway is still inhibited. The cause-effect relationship between SRC abundance and the development of the uLMS ECM subtype is still unclear and needs further analysis.

## 5. Conclusions

Our findings suggest that PSMB9 expression levels can serve as important biomarkers for stratifying uLMS patients that could potentially respond to immune-checkpoint treatment. Further studies using datasets from patient cohorts are necessaire to fully understand the impact of this finding on the patient outcome.

## Figures and Tables

**Figure 1 cancers-14-05007-f001:**
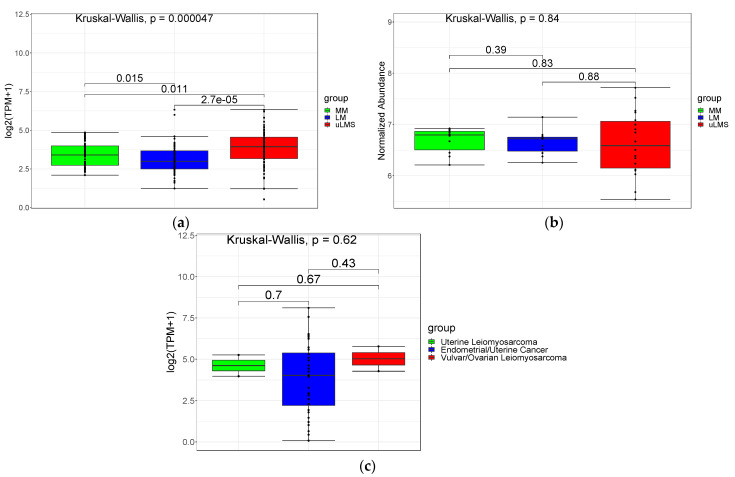
(**a**) PSMB9 expression at gene level between normal adjacent myometrium tissue (MM), leiomyoma (LM) and uterine leiomyosarcoma (uLMS) groups. (**b**) PSMB9 expression at protein level between normal adjacent myometrium tissue (MM), leiomyoma (LM) and uterine leiomyo-sarcoma (uLMS) groups. (**c**) PSMB9 expression at gene level between uterine leiomyosarcoma, endometrial/uterine cancer and vulvar/ovarian leiomyosarcoma cell lines (source: DepMap Portal (https://depmap.org/portal/download/all/, accessed on 10 October 2021).

**Figure 2 cancers-14-05007-f002:**
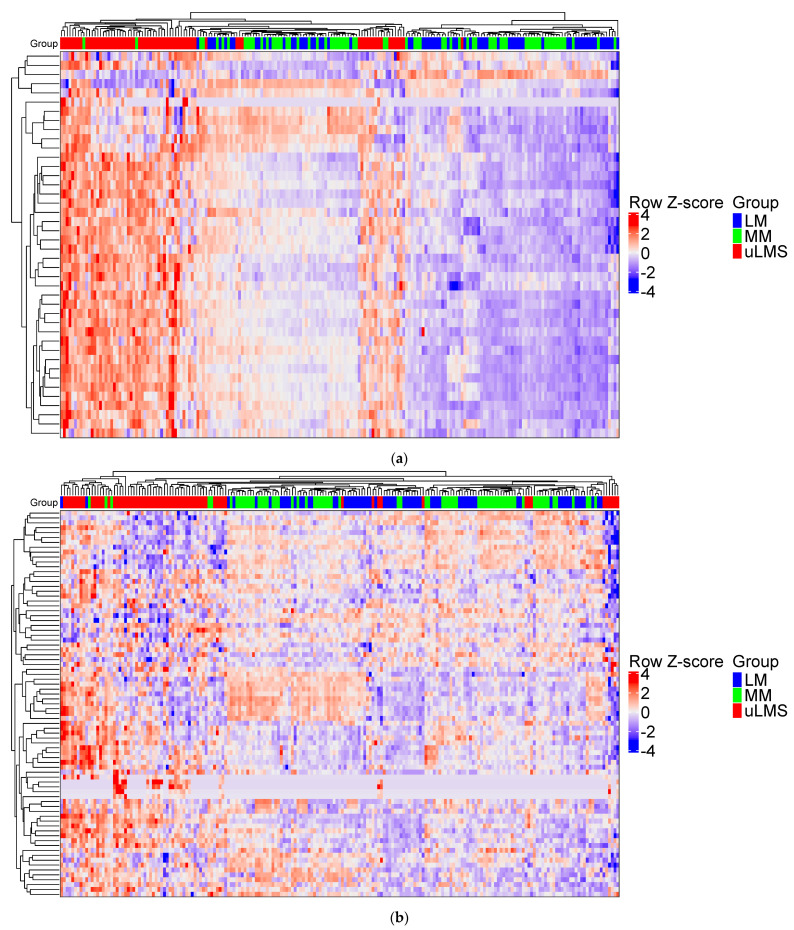
(**a**). Unsupervised clustering based on KEGG PROTEASOME pathway from MSigDB database. (**b**) Unsupervised clustering based on REACTOME INTERFERON GAMMA SIGNALING pathway from MSigDB database as a gene set. LM: Leiomyoma; uLMS: uterine Leiomyosarcoma; MM: myometrium.

**Figure 3 cancers-14-05007-f003:**
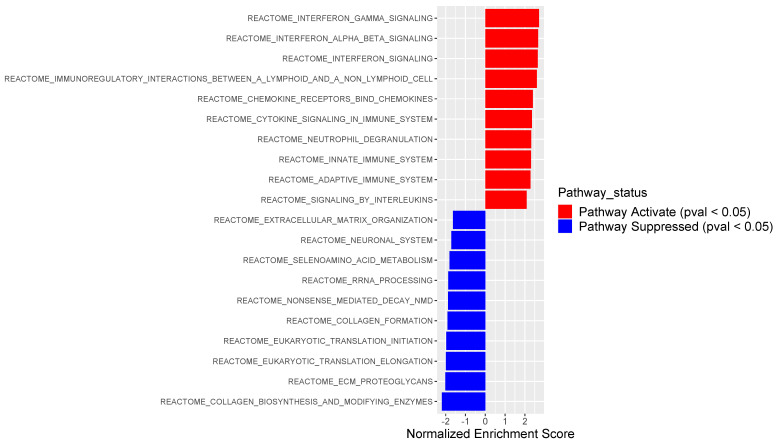
Normalized enrichment scores (NES) of top 10 most activated and top10 most suppressed pathways comparing High versus Low PSMB9 expression level. Red color indicates activation, while blue color indicates suppression. Reactome gene set from MSigDB curated gene set (C2) was used as input. Pathways related to the immune system and inflammatory response were significantly activated, while pathways related to the extracellular matrix were suppressed.

**Figure 4 cancers-14-05007-f004:**
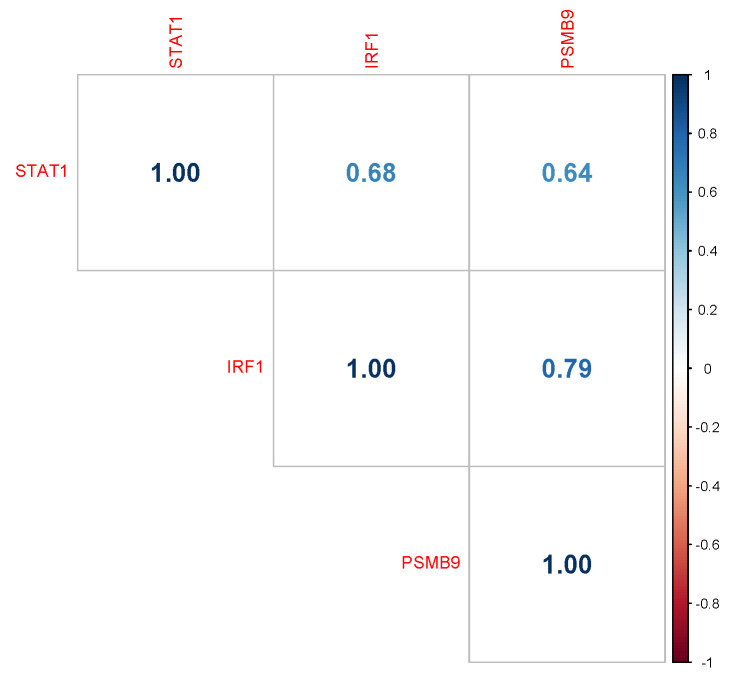
Correlation matrix of genes related to IFNγ signaling through the JAK/STAT1 pathway in all uLMS samples in the 1st and 4th quartiles. All values inside each cell correspond to Pearson correlation coefficient at the 0.05 level.

**Figure 5 cancers-14-05007-f005:**
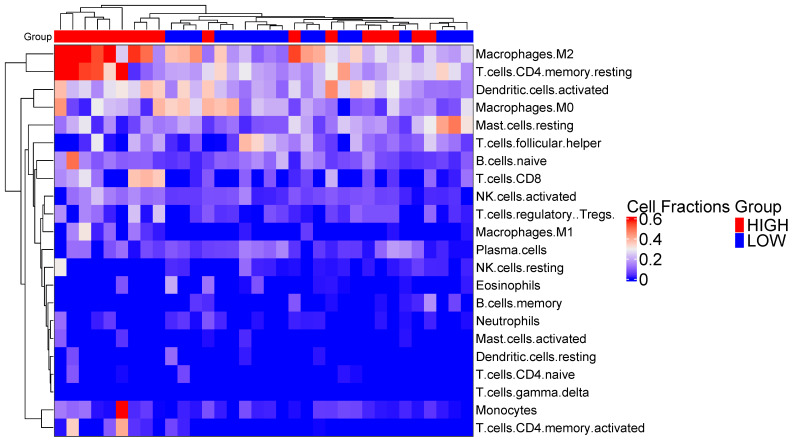
Unsupervised clustering of high and low PSMB9 expression levels based on the absolute cell fractions estimated by CIBERSORTx.

**Figure 6 cancers-14-05007-f006:**
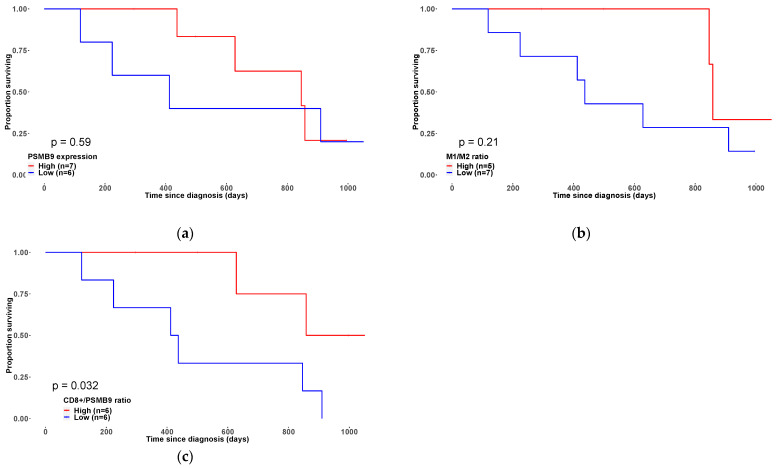
(**a**). 3-year OS of patients grouped based on PSMB9 gene expression *p* = 0.59). (**b**). 3-year OS of patients grouped based on M1/M2 ratio (*p* = 0.21). (**c**) 3-year OS of patients based on CD8+/PSMB9 ratio (*p* = 0.032).

**Figure 7 cancers-14-05007-f007:**
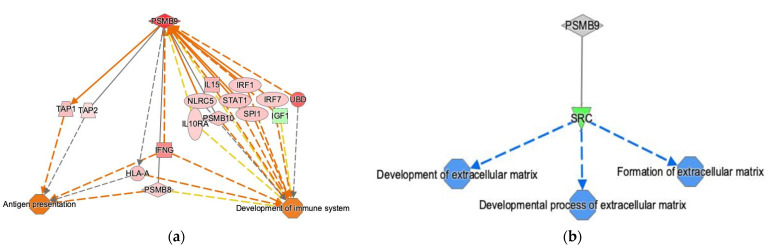
(**a**) Ingenuity Pathway Analysis (IPA) analysis of RNA-Seq data. Prediction of the effects in the high PSMB9 group. High PSMB9 gene expression level leads to activation of “antigen presentation” and the “development of the immune system” through IFNγ. (**b**) IPA analysis of the proteomics data. Prediction of the effects in the PSMB9 group. PSMB9 protein abundance is related to the development and formation of the ECM through SRC. Direct gene interactions are represented by continuous lines, while indirect relationships are represented by dotted lines.

## Data Availability

The data will be shared after the institutional approval.

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
