# Peer review of "The Expression of the Immunoproteasome Subunit PSMB9 Is Related to Distinct Molecular Subtypes of Uterine Leiomyosarcoma"

_cancers, 2022, doi:10.3390/cancers14205007_

Round 1

Reviewer 1 Report

Falcao et al present a paper entitled "The expression of the immunoproteasome subunit PSMB9 is 

related to distinct molecular subtypes of uterine leiomyosarcoma". 

The paper is interesting and focus on a gene that has been poorly studied in LMS so far (3 studies in pubmed).

i have only minor suggestions for the authors that i believe may improve the reading.

- i would add a flowchart showing the different cohort of pts analyzed in the study and describing the type of analysis. 

- some of the figures are very small and are difficult to read (even zooming); in particular fig 1 and 3. please enlarge those

- please delete the section "appendix A" and "Acknowledgments" since are empty

Author Response

Reviewer 1

- I would add a flowchart showing the different cohorts of pts analyzed in the study and describing the type of analysis. 

Answer: this figure was created and added as Supplementary figure 1

- some of the figures are very small and are difficult to read (even zooming); in particular fig 1 and 3. please enlarge those

Answer: the size of the figures was improved.

- please delete the section "appendix A" and "Acknowledgments" since are empty

Answer: we have added information to the acknowledgment session and to appendix A as Supplementary Figure 1A. The whole session Appendix A was added to the correct version, since it contains supplementary figures and tables.

Reviewer 2 Report

The goal of this study was to investigate PSMB9 expression and its relationship with uLMS patient prognosis from molecular data of different tumors sources. Low and high PSMB9 expression levels were used to performe a differential expression analysis and a gene set enrichment analysis. From the statistical analysis, high PSMB9 expression levels mainly present activation of pathways related to IFN-γ and immune system and reduction of development and formation of extracellular matrix. Using CYBERSORTx analysis, high PSMB9 level group showed an improved prognosis related to CD8+/PSMB9 ratio, indicating that PSMB9 expression level may be associated with immune response in uLMS. And a low PSMB9 protein abundance could be related to the inhibition of pathways related to the formation of ECM through the proto-oncogene tyrosine-protein kinase (SRC) gene.

This work is well introduced and well described. Such study could give interesting results / ideas indicating which biological processes are being regulated in the context of uterine leiomyosarcoma.

As a suggestion, I would recommend to the authors to try to present the results with larger plots, allowing the reader to see the results. Here all the figures are so small that it is almost impossible to see anything. In case of figure 2, it can be moved to sup data.

The strategy of the author is very well defined, and the results are well organized and presented. The statistics are appropriate and the conclusions fit well with the results. According to all these remarks, I consider this manuscript as acceptable for publication in Cancers

Author Response

Thanks for your time in reviewing the manuscript.

The size of the figures was improved and we have added figures and tables to the session Appendix A.

Reviewer 3 Report

An intriguing study is the manuscript titled "The expression of the immunoproteasome subunit PSMB9 is associated with distinct molecular subtypes of uterine leiomyosarcoma." As leiomyosarcoma of the uterus is a rare tumor. Using public datasets of RNAseq analysis provides a clear rationale for order to identify potential biomarkers that could be utilized in a novel approach. The manuscript is written well. These few deficiencies exist in the manuscript:-

1. All figures are not clear. Written portions of the figure are not legible.

2. Author mentioned in their mentioned 95 female patients are submitted to surgery but they collected 124 fresh frozen tissues. Please justify?

3. Does author perform any immunohistochemistry of PSMB9 expression on feesh tumor tissue collected?

4. I figure 2 author mentioned z-scores and log2(TPM+1). Both are two different variables. Please justify whether heatmap are made on the bais of Log2 (TPM+1) or z-scores?

Author Response

  1. All figures are not clear. Written portions of the figure are not legible.

Answer: the size of the figures was improved.

  1. Author mentioned in their mentioned 95 female patients are submitted to surgery but they collected 124 fresh frozen tissues. Please justify?

Answer: 95 patients have been submitted to surgery and some patients had more than 1 biospecimen collected, for example, uLMS and MM from the same patient.

  1. Does the author perform any immunohistochemistry of PSMB9 expression on fresh tumor tissue collected?

Answer: we did not perform IHC for PSMB9 in FRESH tissue. However, we have analyzed proteomics data and presented the results in the Figures 1b and 7b. We hope this analysis is sufficient to understand the role of the protein PSMB9 in uLMS.

  1. In figure 2 author mentioned z-scores and log2(TPM+1). Both are two different variables. Please justify whether heatmap are made on the bais of Log2 (TPM+1) or z-scores?

Answer: the row Z-score was calculated based on log2(TPM+1) values. This information was added at the end of the second paragraph of session 4 in Materials and Methods.